# OpenReview forum: "Variational Geometric Information Bottleneck: Toward a Geometric Law of Understanding"
_TMLR — Rejected by TMLR_

### Review · Reviewer_PUjj · 2025-12-01

**Summary Of Contributions:**

The paper proposes the Variational Geometric Information Bottleneck (V-GIB), a framework that unifies mutual-information compression with curvature regularization via tractable geometric proxies. The authors define "understanding" as a trade-off between task-relevant information and geometric simplicity (low curvature and intrinsic dimension). They provide non-asymptotic generalization bounds and introduce "Interpretive Efficiency" as a metric. Experiments are conducted on synthetic manifolds, few-shot tasks, and standard datasets like CIFAR-10.

Strengths:
* The core idea is novel and interesting.
* Theoretical proof seems reasonable.
* The empirical validation of the "curvature-information Pareto frontier" on synthetic and real data provides strong evidence for the proposed trade-off.

Weakness:
1. The theory relies heavily on the assumption that "The data are supported on a compact, smooth d-dimensional Riemannian manifold". In reality, high-dimensional data (like natural images or text) often reside on unions of manifolds with singularities or discrete structures, which the authors admit "may not hold in high-dimensional, non-smooth domains".
2. The paper does not theoretically address how this extends to generative tasks or regression, limiting the claim of a general "Law of Understanding".
3. The authors acknowledge that "Estimating curvature through Hutchinson traces or Jacobian norms introduces non-trivial overhead". However, there is no quantitative analysis of training time and performance gain to justify this cost.
4. The experiments lack validation on large-scale benchmarks, such as ImageNet, and comparisons with large language models.
5. Figure 1 lacks clear labeling for sub-figures (a), (b), (c), and (d), making it difficult to follow the caption.

**Audience:**

Yes

**Audience Explanation:**

The unification of the Information Bottleneck principle with geometric curvature regularization  is a novel conceptual contribution that would interest researchers working on geometric deep learning and information theory.

**Broader Impact Concerns:**

No ethical implications

**Claims And Evidence:**

No

**Claims Explanation:**

The paper claims to establish a "Geometric Law of Understanding" and a general theory of "interpretive efficiency". However, the empirical validation is restricted to small-scale benchmarks (CIFAR-10, Fashion-MNIST)  and synthetic manifolds. The absence of large-scale validation (e.g., ImageNet), diverse tasks and  comparison with large language models makes the claim of a "general law" insufficiently supported by the provided evidence.

**Requested Changes:**

1. Please provide empirical evidence on diverse tasks and large-scale benchmarks to demonstrate the universality of this law.
2. Please provide a quantitative comparison of training time versus performance gain.
3. Please add a dedicated discussion clarifying to which tasks satisfy the theoretical assumptions.
4. Please provide comparisons with Large Language Models.
5. Please polish writing and improve figure details.

---

### Review · Reviewer_GhnC · 2025-12-04

**Summary Of Contributions:**

The paper proposes a unified framework—**Variational Geometric Information Bottleneck (V-GIB)**—that couples mutual-information objectives with geometric regularization, aiming to formalize “structural understanding” as a trade-off between informativeness and curvature-based simplicity. The authors present (1) theoretical generalization bounds depending on intrinsic dimension and curvature; (2) a variational estimator combining MI surrogates with curvature proxies such as Hutchinson traces; (3) several empirical validations on synthetic manifolds, Fashion-MNIST, CIFAR-10, and few-shot scenarios; and (4) an “interpretive efficiency” metric that quantifies information per labeled example.

**Strengths:**
- The idea of integrating explicit curvature regularization into IB-style objectives is interesting and timely.
- The theoretical analysis provides non-asymptotic generalization bounds and curvature-dependent covering-number arguments.
- Experiments are extensive and show consistent geometric–information correlations.

**Weaknesses:**
- Several theoretical results rely on strong smooth-manifold assumptions and bi-Lipschitz encoders, which may not hold in realistic deep networks; the paper does not fully explain when these assumptions fail.
- Empirical demonstrations of curvature effects are suggestive but not fully causal; many plots lack ablations that isolate geometric regularization from generic smoothing or reduced capacity.
- The interpretation claims (e.g., “geometric law of understanding”) are substantially overstated compared with the evidence presented.
- The introduction and presentation are verbose, making it difficult to isolate the key technical innovations.
- Some empirical results appear sensitive to hyperparameters $(\beta,\gamma)$, but the paper does not provide robustness checks or principled tuning guidance.

**Additional Comments:**

- Ambitious and interesting work, but framing often exceeds what is rigorously demonstrated.
- Consolidation of theoretical results and clearer novelty statements would strengthen the paper.
- Experiments are broad but need stronger causal and baseline analyses.
- The core idea—integrating curvature with information bottleneck objectives—has potential and could make a solid contribution with further refinement.

**Audience:**

Yes

**Audience Explanation:**

The intersection of geometric regularization, information bottlenecks, and representation learning is timely and relevant. However, the manuscript's overstated narrative may limit its appeal to readers seeking precise and rigorously validated contributions.

**Broader Impact Concerns:**

No major ethical risks are inherent to the method, but the manuscript’s claims about “human-aligned understanding” could be misinterpreted. Curvature-smooth embeddings do not ensure causal or semantic interpretability, and small-scale human studies do not validate reliability in high-stakes domains. A Broader Impact Statement should clarify these limitations and caution against over-interpreting geometric regularity as genuine conceptual alignment.

**Claims And Evidence:**

No

**Claims Explanation:**

1. Experiments show correlation but not causation; missing baselines that control for capacity, smoothing, or architectural effects.

2. The assumptions (smooth manifolds, bounded curvature, bi-Lipschitz encoders) rarely hold for ReLU networks; no justification for practical validity.

3. Human-alignment assertions are overstated; the evaluation is small-scale and lacks comparative baselines.

4. Curvature and MI estimators are noisy; variance diagnostics are limited.

Overall, evidence is suggestive but not sufficiently convincing to support the strongest claims.

**Requested Changes:**

**Critical:**
1. Provide causal evidence isolating curvature’s contribution; add capacity- and smoothness-controlled baselines.
2. Clarify and justify manifold and bi-Lipschitz assumptions; discuss practical violations in deep networks.
3. Tone down speculative framing (“geometric law of understanding”) and limit claims to what is empirically supported.
4. Add estimator variance diagnostics and sensitivity analyses for curvature and MI estimators.
5. Streamline exposition and clearly delineate novel contributions relative to IB and manifold regularization.

**Recommended:**
- Compare with alternative geometric regularizers (e.g., Laplacian, Hessian penalties).
- Provide practical tuning guidelines for $\beta$ and $\gamma$.
- Improve figure clarity and tighten experimental narrative.

---

### Review · Reviewer_HUMq · 2025-12-04

**Summary Of Contributions:**

The paper proposes the Variational Geometric Information Bottleneck (V-GIB), a unified framework that modifies the standard Information Bottleneck (IB) objective by adding geometric regularization terms, specifically penalizing curvature and intrinsic dimensionality. The authors derive theoretical generalization bounds linking these geometric properties to sample efficiency. They aim to validate these claims through synthetic manifold recovery, few-shot learning tasks, and standard image classification benchmarks, arguing for a "geometric law of understanding" where smoothness substitutes for data volume.

**Additional Comments:**

This paper will require major revision before any meaningful review, let along publication. I highly suspect the current paper was generated by an LLM.

**Audience:**

No

**Audience Explanation:**

The core ideas are likely to interest several sub-communities. But the current state of the paper and the current evidence in the paper will not result in any audience in the community.

**Claims And Evidence:**

No

**Claims Explanation:**

While the theoretical analysis seems plausible, the empirical validation is critically flawed due to unfair baselines and missing results.
 The evidence supporting the claim that V-GIB outperforms standard methods is insufficient due to severe discrepancies between the experimental protocol (Section 3) and the reported results (Section 4).

* Section 3.2 explicitly states that baselines include "vanilla ResNet, VIB... and manifold-regularized ResNet". However, Section 4.4 and Tables 1 & 3 report results only for V-GIB. No comparative accuracy numbers are provided, making it impossible to verify if V-GIB actually outperforms a standard ResNet.
* The protocol promises results on "Omniglot, miniImageNet, and tieredImageNet" comparing against MAML and Matching Networks. These results are completely absent from the Experimental Validation section.
* Section 3.3 mentions comparing against "Transfer-learning baselines (ResNet and CLIP encoders)". These comparisons are missing.
* In Section 4.3, the paper claims V-GIB outperforms a baseline because of "geometric structure, not model size". However, the baseline is described as a "random linear baseline", while V-GIB uses a non-linear encoder with "two hidden layers (128 ReLU units)". Comparing a linear model to a non-linear neural network on a non-linear manifold task (Swiss Roll) is an unfair comparison that invalidates the specific claim regarding model capacity.

**Requested Changes:**

* The authors must populate the results for the baselines explicitly listed in the Experimental Protocol (Section 3).
* The comparison in Section 4.3 against a "random linear baseline" is methodologically invalid for a non-linear manifold task. Replace or augment this with a fair baseline.
* The clarity of the paper is poor, a significant rewrite is needed.

---

### Review · Reviewer_qKda · 2025-12-04

**Summary Of Contributions:**

Authors introduce a framework, called variational information geometric bottleneck.
The proposed metric balances the informativeness of the encoder, by computing the mutual information with the target labels, and geometry, by regularizing towards smoothed low-dimensional embeddings.
The work argues that this metric is a suitable proxy for controlling the generalization error, and that encoders with curvature-aware regularization leads to more data-efficient methods.
Furthermore, an interpretive efficiency is introduced, which quantifies the per-label effect of the samples.

**Audience:**

Yes

**Audience Explanation:**

The authors aim to provide a theory and practical results for the data-efficient learning, which is an important question.
Authors conduct diverse experiments to test their hypotheses and claims.
These experiments support the claim that the curvature-aware encoders able to learn efficiently from a small amounts of data.
However, I believe that the clarity of some contributions, in particularly theoretical, is currently insufficient.

**Broader Impact Concerns:**

I do not have any concerns regarding the ethical implications of the work.

**Claims And Evidence:**

No

**Claims Explanation:**

The clarity of the exposition needs to be significantly improved.
Many quantities in the main text are used without a proper introduction, and there is no explanations in the appendix.
For example:

1. Definition of the covering number $\mathcal{N}$ is missing,
2. Definition of the risk function $R(f)$,
3. Definition $I(\phi)$ in Proposition 2.11,
4. Definition of the reach and sectional curvature in the context of Riemannian manifolds,
5. Definition of the alignment mutual information,
6. In Table 4, two functions, $C(\phi)$ and $\mathcal{C}(\phi)$ appear with reference to the main text. In the main text, $C(\phi)$ does not appear,
7. In Table 4, $U(\phi)$ is defined through $C(\phi)$, which disagrees with the definition in the abstract and the main text.

Furthermore, the current state of the theoretical contributions is just a collection of Propositions and Theorems in Sections 2.4 and 2.5 with only a brief explanation in Section 2.6 of how they are connected.
What is the main result that highlights the importance of the defined quantity remains vague.

"the encoder class is equipped with a curvature regularizer of weight $\beta$" could the author please clarify what this sentence means?

"Shape of understanding" in the conclusion is a very vague claim.

There is limited discussion of the computational efficiency of computing the proposed quantity.
In the experiments, it seems that the KL divergence is used as a proxy for the mutual information.
How is this related to the discussion in Appendix A.5?

No discussion regarding the bi-Lipschitzness assumption. Do we expect it to hold for interesting examples?

**Requested Changes:**

I suggest to improve the clarity of the theoretical results and their connection to the practical experiments. Please see above most of my concerns.

Also, some results currently appear to only contain proof sketches, not full proofs (Theorem 2.4, Theorem A.2).

Furthermore, Theorems 2.8 and 2.10 should probably be joined, as their statements and proofs are very related.

---

### Review · Reviewer_uLeC · 2025-12-05

**Summary Of Contributions:**

This paper proposes Variational Geometric Information Bottleneck (V-GIB), a novel geometric extension of information bottleneck that defines learning as a trade-off between predictive information and geometric simplicity. By adding explicit penalties for curvature and intrinsic dimension to the Information Bottleneck objective, the authors theoretically prove that generalization is governed by the smoothness of the learned manifold and empirically demonstrate that enforcing low-curvature geometry largely improves sample efficiency and interpretability.

**Audience:**

Yes

**Audience Explanation:**

- The paper presents a detailed, grounded framework connecting the Information Bottleneck with geometric simplicity, offering an interesting perspective on the learning process of modern models
 - This paper theoretically justifies why smooth representations generalize well, offering non-asymptotic sample complexity bounds which prove that generalization error scales with the data's intrinsic dimension ($d$) and that curvature ($\kappa$) acts as a stabilizing capacity term.

**Claims And Evidence:**

No

**Claims Explanation:**

- The reliance on Hutchinson trace estimators for curvature requires Hessian-vector products, introducing significant computational cost compared to standard VIB. Although the authors acknowledge this in the limitations, I suggest the authors provide a detailed GPU-hour or Wall-clock time comparison for each method to quantify the practical overhead.
- While the theory advocates for _minimizing_ curvature and Section 4.4 explicitly states that *"high accuracy coincides with lower-curvature"*, the experimental results in Figure 2(d) contradict this by showing a strong positive correlation ($r \approx 0.96$) between accuracy and curvature energy. Can author further clarify this inconsistency?
- The reliance on mini-batch covariance for $\hat{d}_{PR}$ 1 creates mathematical instability, as the matrix rank is artificially capped by the batch size. Consequently, the geometric penalty fluctuates based on optimization hyperparameters rather than the true manifold structure, potentially yielding incorrect gradients. The authors should provide discussion or ablation studies regarding the estimator's sensitivity to batch size to address this flaw.
- Table 1 reports metrics exclusively for V-GIB, omitting the corresponding baseline results. Similarly, baseline metrics are missing for the Fashion-MNIST dataset. Without these baselines, it is difficult to verify the claimed performance gains.
- Currently, V-GIB are mainly validated on ResNets and small MLPs; it remains untested on Transformers. I wonder whether attention will possess different geometric properties.
- While the authors acknowledge the small sample size ($m=6$) of the human study as a limitation, this constraint appears readily mitigatable with current technology. It would be interesting to investigate if LLMs/VLMs can automatically scale this process.

**Requested Changes:**

Please see the weakness above for the detailed information.

---

### Review · Reviewer_vU3b · 2025-12-06

**Summary Of Contributions:**

The paper proposes Variational Geometric Information Bottleneck (V-GIB), which tries to model understanding as a trade-off between task-relevant mutual information and a geometric complexity penalty on the encoder. Concretely, it defines an objective mixing curvature-like and intrinsic-dimension terms, and introduces interpretive efficiency as a performance-per-complexity measure. The authors also give generalization bounds on manifolds and theorems connecting information, geometry and alignment. Experiments are conducted on synthetic data, Fashion-MNIST and CIFAR-10.

**Audience:**

Yes

**Audience Explanation:**

The paper studies the intersection of information bottleneck, geometric / manifold-based learning, and interpretability, which is interesting. A framework that explicitly ties intrinsic dimension, curvature and information retention is conceptually appealing and may spark useful discussion.

**Claims And Evidence:**

No

**Claims Explanation:**

In my view, the evidence does not yet match the claims:

- On the theory side, the generalization bounds lean heavily on standard manifold assumptions and covering-number arguments. The new curvature-dependent terms are only sketched, and there is no solid proof that the proposed curvature penalty on the encoder actually controls the geometric quantities that appear in the bounds.

- The paper blurs different notions of geometry: it assumes a curved data manifold, penalizes encoder Hessians, and then interprets empirical 'curvature energy' as if all of these were directly linked. That connection is asserted without clear evidence.

- On the experimental side, many of the claims including few-shot gains, human alignment are mentioned but **results are not presented**, and there is no comparison to simpler baselines like standard IB or existing Jacobian/curvature regularization.

**Requested Changes:**

- Please clearly distinguish the data-manifold curvature, representation-manifold curvature, and encoder-Hessian penalties. Please either prove how the proposed regularizer controls the geometric quantities in the bounds.

- Please provide more evidence for statements like geometric laws of understanding and strong human alignment.

- For the few-shot and low-data tasks, please report full results and compare against additional baselines, e.g., VIB, IB + standard Jacobian/curvature regularization. For CIFAR-10/Fashion-MNIST, please include additional ablations to identify what part of V-GIB actually helps.

---

### Decision · Action_Editor_fCGV · 2026-02-11

**Recommendation:** Reject

**Additional Comments:**

The paper received many concerns from all the reviewers, especially in terms of the claim that lack both theoretical and experiment supports. The AE agreed and the authors may consider to significantly rewrite the paper or prepare a new submission. A major revision as resubmission may not apply.

**Audience:**

Yes

**Audience Explanation:**

the paper addresses relevant and important topics in machine learning, and provides useful insights.

**Claims And Evidence:**

No

**Claims Explanation:**

As the reviewers' comments, the paper needs more theoretical and empirical studies to justify the claiming points.

**Resubmission Of Major Revision:**

The authors may consider submitting a major revision at a later time.